# Bayesian Optimistic Optimization: Optimistic Exploration for Model-based Reinforcement Learning

**Chenyang Wu**[1], **Tianci Li**[1], **Zongzhang Zhang**[1,*], **Yang Yu**[1,2]

[1]National Key Laboratory for Novel Software Technology, Nanjing University, Nanjing, China
[2]Pazhou Lab, Guangzhou, China
`{wucy, litc}@lamda.nju.edu.cn, {zzzhang, yuy}@nju.edu.cn`

## Abstract

Reinforcement learning (RL) is a general framework for modeling sequential decision making problems, at the core of which lies the dilemma of exploitation and exploration. An agent failing to explore systematically will inevitably fail to learn efficiently. Optimism in the face of uncertainty (OFU) is a conventionally successful strategy for efficient exploration. An agent following the OFU principle explores actively and efficiently. However, when applied to model-based RL, it involves specifying a confidence set of the underlying model and solving a series of nonlinear constrained optimization, which can be computationally intractable. This paper proposes an algorithm, Bayesian optimistic optimization (BOO), which adopts a dynamic weighting technique for enforcing the constraint rather than explicitly solving a constrained optimization problem. BOO is a general algorithm proved to be sample-efficient for models in a finite-dimensional reproducing kernel Hilbert space. We also develop techniques for effective optimization and show through some simulation experiments that BOO is competitive with the existing algorithms.

## 1 Introduction

Reinforcement learning (RL) is a sequential decision-making problem in which an agent acts in an unknown environment while maximizing the cumulative rewards it receives [1, 2]. In this paper, we consider the RL in Markov decision processes (MDPs), where the agent observes the state of the environment at each timestep and makes decisions accordingly. Since the environment is unknown, maximizing the cumulative rewards naturally involves a trade-off between exploration and exploitation. Exploitation is to make the best-rewarding decision based on the agent's information, and exploration means actively gathering information about the environment so that the agent understands better about the environment and thus makes better decisions in the future. An algorithm cannot be sample-efficient without balancing them properly.

Theoretically, the exploration and exploitation dilemma admits a Bayesian optimal solution [3]. That is to consider the RL problem a so-called Bayes-Adaptive MDP (BAMDP), a special case of partially observable Markov decision processes (POMDPs), where the parameter of the dynamics is unobservable. Although this formulation provides useful insights, the POMDP formulation is computationally intractable [4–6]. Therefore, all practical algorithms [7–10] resolve this dilemma by achieving a delicate balance between seeking rewards and gathering information. Optimism in the face of uncertainty (OFU) is one of the conventionally successful approaches for this balance and is established as an efficient learning principle in various cases [11–13].

---

[*]Corresponding Author

36th Conference on Neural Information Processing Systems (NeurIPS 2022).

The OFU principle makes the agent an optimist who is always optimistic about the uncertainty of the environment. If it ever stands a chance that a policy is highly profitable, the agent will try it out. By executing an optimistic policy, the agent either receives high rewards or gains information after observing unexpected outcomes. OFU involves constructing a confidence set of possible MDPs and solving for the most optimistic one within the confidence set. Unfortunately, this is typically a constrained nonlinear non-concave optimization and is impractical to solve [14, 15]. Therefore, a model-based OFU algorithm is previously deemed an inferior choice [15].

In this paper, we propose a model-based OFU algorithm, Bayesian optimistic optimization (BOO), which enforces the constraint via a dynamic weighting technique and therefore makes the optimization more practical. We show that BOO is a general-purpose model-based OFU RL algorithm that is both provably sample-efficient and comparatively computationally tractable. Our contribution is three-fold:

- We extend the OFU algorithms to the Bayesian setting by requiring the model to reside in a Bayesian credible region. BOO is then proposed as the Lagrangian relaxation of this constrained optimization.
- We prove that BOO is efficient in terms of the frequentist regret for a finite-dimensional reproducing kernel Hilbert space (RKHS).
- We derive optimization methods for BOO and demonstrate empirical evidence that BOO is competitive with UCRL2 [16] and PSRL [17].

## 2 Preliminaries

This section introduces the preliminaries, including RL in MDPs and the principle of optimism in the face of uncertainty.

### 2.1 Reinforcement Learning in Markov Decision Processes

We consider the problem where a reward-seeking agent repeatedly interacts with a finite-horizon MDPs $M = (\mathcal{S}, \mathcal{A}, P, R, H, s_1)$, where $\mathcal{S}$ and $\mathcal{A}$ are the state and action spaces, respectively. The algorithm adopted by the agent is denoted as $\mathfrak{A}$. At each episode, the agent is spawned at the initial state $s_1 \in \mathcal{S}$. It takes an action $a_h \in \mathcal{A}$ at each period $h$ within an episode and receives a random reward, $r_h \sim R(s_h, a_h)$. Then, the environmental state transitions to the next state, $s_{h+1} \sim P(s_h, a_h)$. This process repeats until the episode ends at the $H$-th period, and another episode begins. Note that the fixed initial state is not a stringent condition since starting from an initial distribution $\rho_0$ is equivalent to starting from a special state $s_1$ such that $P(s|s_1, a) = \rho_0(s)$ holds for any state $s$ and action $a$. A finite MDP is an MDP with finite state and action spaces.

A policy $\pi$ is a function mapping a state and a period to an action distribution. The value function for the MDP $M$, policy $\pi$, and period $h$ is defined recursively as

$$
\begin{aligned}
V_h^{\pi,M}(s_h) &= \mathbb{E}_{a_h \sim \pi(s_h, h)} \left[ Q_h^{\pi,M}(s_h, a_h) \right], \forall h \in [H], \\
Q_h^{\pi,M}(s_h, a_h) &= \bar{R}^M(s_h, a_h) + \mathbb{E}_{s_{h+1} \sim P^M(s_h, a_h)} \left[ V_{h+1}^{\pi,M}(s_{h+1}) \right], \forall h \in [H-1],
\end{aligned}
\tag{1}
$$

where $\bar{R}^M(s, a) = \mathbb{E}_{r \sim R^M(s, a)}[r]$, and $Q_H^{\pi,M}(s_H, a_H) = \bar{R}^M(s_H, a_H)$.

We use $X_{k,h}$ to represent the variable $X$ at the $h$-th period of the $k$-th episode. For notational convenience, $X_{k,h}$ is sometimes abbreviated as $X_{kh}$. The history prior to $k$-th episode is defined as $\mathcal{H}_k = (s_{1,1}, a_{1,1}, r_{1,1}, s_{1,2}, \ldots, s_{k-1,H}, a_{k-1,H}, r_{k-1,H})$. The agent is assumed to be capable of memorizing the entire history. In this paper, we consider model-based RL algorithms which produce a model $M_k$ per episode in light of the history $\mathcal{H}_k$ and derive its corresponding optimal policy $\pi_k \in \arg\max_\pi V_1^{\pi,M_k}(s_1)$ for execution. We use the terms model and MDP interchangeably. For any history $\mathcal{H}$, $\mathfrak{A}(\mathcal{H})$ defines a distribution over models and policies. Within the $k$-th episode, the agent samples an action from $\pi_k(s_{kh}, h)$ at each period $h$.

In the frequentist viewpoint, there exists an unknown true MDP $M^*$. We abbreviate $V_h^{\pi^*,M^*}$ as $V_h^*$, where $\pi^*$ is an optimal policy of the true MDP $M^*$. The frequentist performance metric, regret, is

defined in terms of the true MDP:

$$\text{Regret}(T, \mathfrak{A}, M^*) = \mathbb{E}_{\mathcal{H}_{K+1} \sim \mathfrak{A}, M^*} \left[ \sum_{k=1}^{K} \Delta_k \right], \tag{2}$$

where $\Delta_k$ is defined as $\mathbb{E}_{\pi_k \sim \mathfrak{A}(\mathcal{H}_k)} \left[ V_1^*(s_1) - V_1^{\pi_k, M^*}(s_1) \right]$, $K$ is the total number of episodes, $T = KH$ is the number of the total time steps, and $\mathcal{H}_{K+1} \sim \mathfrak{A}, M^*$ means that the history is sampled by the interaction of the algorithm $\mathfrak{A}$ and the real MDP $M^*$. In the Bayesian viewpoint, the unknown MDP $M^*$ is treated as a random variable and assigned a prior $\rho_M$. All MDPs in the support of $\rho_M$ differ only in the transition function $P$ and the reward function $R$. The Bayesian objective of the agent is to minimize the Bayesian regret up to time $T$, $\text{BayesRegret}(T, \mathfrak{A}, \rho_M) = \mathbb{E}_{M^* \sim \rho_M} [\text{Regret}(T, \mathfrak{A}, M^*)]$.

## 2.2 Optimism in the Face of Uncertainty

Optimism in the face of uncertainty is a strategy for information gathering. When the optimal action is not clear given the current information, it is preferable to hazard an optimistic guess. If we make an atrocious guess, we effectively rule that out and pick another next time. Otherwise, we end up finding a competitive solution that incurs little regrets. This idea is mathematically realized as a constrained optimistic optimization, $\max_{\pi_k, M_k} V_1^k(s_1)$ s.t. $M_k \in \mathcal{M}_k$, where $\mathcal{M}_k$ is a confidence set constructed using empirical data $\mathcal{H}_k$ such that $M^* \in \mathcal{M}_k$ with high probability. The pseudocode of the OFU algorithm is shown in Algorithm 1.

---

**Algorithm 1** OFU RL

---

1: **for** episode $k = 1, 2, \ldots$ **do**
2:     Construct a confidence set $\mathcal{M}_k$ with $\mathcal{H}_k$
3:     Compute $\pi_k \in \arg\max_{\pi} \max_{M_k} V_1^{\pi, M_k}(s_1)$ s.t. $M_k \in \mathcal{M}_k$
4:     Execute $\pi_k$ for an episode

---

# 3 Related Work

One line of research resolves the exploration and exploitation dilemma by formalizing the RL problem as a planning problem in Bayes-Adaptive MDPs (BAMDPs) [3], which treats the unknown MDP parameter as an additional hidden variable and maintains a belief distribution of the parameter. This line of work shares the scalability problem, which is caused by the exponential increase of possible histories w.r.t. the planning horizon. That is, the planning in BAMDPs is PSPACE-complete [4] and requires exponential time to solve. Intuitively, this is because methods based on BAMDPs deal with the belief distributions of MDPs (or histories) rather than a single MDP. Efficient planning algorithms in BAMDPs do exist. Exploiting the root sampling technique, we can implement an algorithm that gets rid of the posterior distribution and only requires posterior sampling [18]. Nonetheless, the underlying scalability issue remains prominent.

Methods following the OFU principle construct an optimistic estimate of the unknown MDP and execute its optimal policy. Since these methods plan on a single MDP estimate, they are computationally preferable compared to methods based on BAMDPs. The UCRL2 [16] is one such method. However, unlike tabular and linear MDPs for which constructing an optimistic estimate is analytically tractable [11, 13], constructing an optimistic estimate for general MDPs involves a constrained joint optimization of model and policy and is computationally prohibitive. Hence, previous model-based OFU algorithms [19, 20] for general model classes cannot be implemented and rely on posterior sampling for exploration.

The posterior sampling for reinforcement learning (PSRL) works by selecting a random MDP from the posterior distribution and executing its optimal policy [21]. This strategy ensures that a policy is selected according to the probability that it is the optimal policy of the real model. It is shown that PSRL is at least as good as any frequentist OFU algorithm in terms of Bayesian regret [22, 15]. PSRL is argued to be better than optimism since it is more computationally tractable [15]. Nevertheless, as opposed to OFU algorithms, incremental implementation of PSRL is challenging because it requires replanning after each sampling. Practical implementation instead tries to directly sample from the

posterior distribution of the optimal value function [23–25]. However, the theoretical guarantee for these methods is only established for tabular MDPs [25].

H-UCRL was devoted to resolving the intractability of model-based optimistic exploration for general models [26]. It proposes to convert the joint optimization of model and policy into a hallucinated control problem. This approach ignores the correlation between state-action pairs and treats them separately, causing inefficiency as reflected by the extra dependency on the cumulative posterior variance in their regret bound. Our method, BOO, also tries to attack the intractability of optimism, which builds an optimistic model by optimizing both the value and the log-posterior density. It avoids the defect of H-UCRL since maximizing the log-posterior density naturally enforces the correlation between state-action pairs.

We note that, when the prior is uniform, this idea is equivalent to balance value versus log-likelihood, which is first explored in [27] and is named as reward-biased maximum likelihood estimation (RBMLE). They have applied this approach to multi-armed bandits [28], contextual bandits [29], and RL where the model belongs to a known finite set [27, 30]. A constrained version of RBMLE is also successfully applied to RL of linear quadratic control systems [31]. Our algorithm can be considered generalizing RBMLE to a Bayesian perspective and finite dimensional RKHS. In this regard, BOO could also be referred to as reward-biased maximum a posteriori.

Concerning the regret analysis, previous regret analysis for general model-based RL [19, 20] relies on the fact that the constructed model is the most optimistic one in the confidence set. The regret analysis of BOO differs with them significantly since the model constructed by BOO may not belong to a confidence set or a credible region. This difference entails a distinct analysis, where we show that neither the large deviation from the real model nor the possibly pessimistic estimation of the model causes a large regret.

## 4 Bayesian Optimistic Optimization

In this section, we derive the learning objective of Bayesian optimistic optimization (BOO) as a Lagrangian relaxation of a constrained optimization problem and give an intuitive interpretation of the resulting objective. Assuming the model class resides in a finite-dimensional RKHS, we show that BOO enjoys $\tilde{O}(\sqrt{K})$ regret.

### 4.1 Constrained BOO

The conventional OFU algorithm, as demonstrated in Algorithm 1, contains a constrained optimistic optimization, where we look for an optimistic MDP $M_k \in \mathcal{M}_k$ and its corresponding optimal policy such that the value is maximized. The constrained BOO is almost the same (see Algorithm 2), except the frequentist confidence set is now replaced with the Bayesian credible region. A credible region with a $1 - \alpha_k$ level of confidence is a set $\mathcal{M}_k$ such that $\Pr(\mathcal{M}_k|\mathcal{H}_k) \geq 1 - \alpha_k$, where $\Pr(\cdot|\mathcal{H}_k)$ is the posterior distribution given history $\mathcal{H}_k$, and $\Pr(\mathcal{M}_k|\mathcal{H}_k) = \int_{M_k \in \mathcal{M}_k} \Pr(M_k|\mathcal{H}_k) \, \mathrm{d}M_k$.

By the construction of the credible region, we have $M^* \in \mathcal{M}_k$ holds with probability $1 - \alpha_k$ given any history $\mathcal{H}_k$. Therefore, the per-episode Bayesian regret is bounded with probability $1 - \alpha_k$,

$$\mathbb{E}[\Delta_k] \leq \underbrace{\mathbb{E}\left[V_1^*(s_1) - V_1^k(s_1) \big| M^* \in \mathcal{M}_k\right]}_{\tilde{\Delta}_k^{\mathrm{opt}}} + \underbrace{\mathbb{E}\left[V_1^k(s_1) - V_1^{\pi_k, M^*}(s_1) \Big| M^* \in \mathcal{M}_k\right]}_{\tilde{\Delta}_k^{\mathrm{conc}}} \leq \tilde{\Delta}_k^{\mathrm{conc}},$$

(3)

where $V_h^k = V_h^{\pi_k, M_k}$, and the optimism term $\tilde{\Delta}_k^{\mathrm{opt}}$ is less than or equal to 0 by construction. We define a distance metric $d(M_1, M_2) = \max_\pi |V_1^{\pi, M_1}(s_1) - V_1^{\pi, M_2}(s_1)|$. It is preferable to have a credible region $\mathcal{M}$ such that the set width $\max_{M_1, M_2 \in \mathcal{M}} d(M_1, M_2)$ is minimized since the set width certifies an upper bound on $\tilde{\Delta}_k^{\mathrm{conc}}$. However, designing a value concentration credible region could be intractable for generic model classes. As a reasonable alternative, we propose the highest density region (HDR) [32] or, in the Bayesian context, the highest posterior density (HPD) region [33], i.e., $\mathcal{M}_k = \{M_k | \Pr(M_k|\mathcal{H}_k) \geq \epsilon_k\}$, where $\epsilon_k$ is the largest constant such that $\Pr(\mathcal{M}_k|\mathcal{H}_k) \geq 1 - \alpha_k$. This kind of region features a desirable property that it occupies the smallest volume in the sample space among all credible regions of the same confidence level and has a potentially small set width.

There are two problems preventing the constrained BOO from being a practical algorithm. The first is that the $\epsilon_k$ in the definition of the HPD region is unknown. Although we may approximate it with the Monte Carlo approximation [34], an estimator of high/low quantiles has high variance rendering the approximation difficult. The other difficulty is that the constrained joint optimization of model and policy is an NP-hard problem even in bandits with linear reward and quadratic constraints [14].

---

**Algorithm 2** Constrained BOO

---

1: **for** episode $k = 1, 2, \ldots$ **do**
2:    Construct a **credible region** $\mathcal{M}_k$ with $\mathcal{H}_k$
3:    Compute $\pi_k \in \arg\max_\pi \max_{M_k} V_1^{\pi, M_k}(s_1)$   s.t.   $M_k \in \mathcal{M}_k$
4:    Execute $\pi_k$ for an episode

---

### 4.2 BOO as Lagrangian Relaxation of Constrained BOO

By introducing a Lagrange multiplier $\lambda_k$, we transform the constrained BOO with HPD regions into an unconstrained optimization problem, $\max_{\pi, M} \left( V_1^{\pi, M}(s_1) + \lambda_k(\log \Pr(M|\mathcal{H}_k) - \log \epsilon_k) \right)$, where $\Pr(M|\mathcal{H}_k) = \frac{\Pr(\mathcal{H}_k|M)\Pr(M)}{\Pr(\mathcal{H}_k)}$. Once the Lagrange multiplier is determined, the constant $\log \epsilon_k$ and the marginal likelihood $\log \Pr(\mathcal{H}_k)$ are irrelevant, and the optimization is equivalent to $\max_{\pi, M} \left( V_1^{\pi, M}(s_1) + \lambda_k(\log \Pr(\mathcal{H}_k|M) + \log \Pr(M)) \right)$. This gives rise to the BOO algorithm as shown in Algorithm 3.

---

**Algorithm 3** BOO

---

1: **for** episode $k = 1, 2, \ldots$ **do**
2:    Compute $\pi_k \in \arg\max_\pi \max_M \left( V_1^{\pi, M}(s_1) + \lambda_k(\log \Pr(\mathcal{H}_k|M) + \log \Pr(M)) \right)$
3:    Execute $\pi_k$ for an episode

---

However, the problem is how to determine the value of $\lambda_k$. We note that the following argument by posterior sampling provides an inexact yet insightful viewpoint. A strict discussion is presented in Appendix B. The maximization of the BOO objective can be considered the problem of selecting the best one among multiple posterior samples (see Algorithm 4). When only one sample is taken, Algorithm 4 reduces to the well-known PSRL [21]. As the number of samples $j$ goes to infinity, maximizing among all posterior samples gives approximately the solution of the above optimization.

---

**Algorithm 4** BOO via Posterior Sampling

---

1: **for** episode $k = 1, 2, \ldots$ **do**
2:    Sample $M_k^1, M_k^2, \ldots, M_k^j \sim \Pr(\cdot|\mathcal{H}_k)$
3:    Compute $\pi_k \in \arg\max_\pi \max_{i \in [j]} \left( V_1^{\pi, M_k^i}(s_1) + \lambda_k(\log \Pr(\mathcal{H}_k|M_k^i) + \log \Pr(M_k^i)) \right)$
4:    Execute $\pi_k$ for an episode

---

Notice that we can regard Algorithm 4 as maximizing over several identically distributed random variables, $\lambda_k \log \Pr(M_k^i|\mathcal{H}_k) + V_1^{\pi_k^i, M_k^i}(s_1)$. The magnitude of their mean does not matter since subtracting a constant will not change the optimum. The thing that matters is their variation. We need to make sure that the variation is not dominated by either $V_1^{\pi_k^i, M_k^i}(s_1)$ or $\lambda_k \log \Pr(M_k^i|\mathcal{H}_k)$. If the value dominates the variation, the algorithm will select an unreliable model, which causes inefficiency. If the variation is dominated by the probability, it shows a strong preference for high-probability models and hesitates to explore.

Theorem 4.1 shows that the standard deviation of log probability is at least a constant (proof in Appendix A).

**Theorem 4.1** (Variation of the log-posterior density). *Suppose $X^n = (X_1, X_2, \ldots, X_n)$ are observations from a stochastic process whose distribution $P_\theta$ depends on $\theta \in \Theta$, an open subset of $\mathbb{R}^m$. Assume that the posterior is asymptotic normal, and the log-posterior density is continuous. The variance of the log-posterior density satisfies $\liminf_{n \to \infty} \text{Var}_\theta [\log \Pr(\theta|X^n)] \geq m/2$.*

The process where an algorithm $\mathfrak{A}$ interacts with a random MDP parameterized by $\theta$ is a stochastic process where the observation $X_t$ is the state-action pair $(s_t, a_t)$, and the observation distribution $\Pr(X^n|\theta, \mathfrak{A})$ is a distribution depending on $\theta$. Thus, by Theorem 4.1, we know that if the posterior distribution is asymptotic normal, then, for a random sample $\theta$ from the posterior distribution, the variance of the random variable $\log \Pr(\theta|X^n)$ is at least $m/2$ when $n$ is large enough.

Theorem 4.1 relies on the asymptotic normality of the posterior distribution, which is satisfied under some regularity conditions specified in [35]. The theorem only certifies a lower bound on the variance rather than establishing its convergence. Arguably, this is mostly a technical issue arising from the unboundedness of the log-posterior density. It is possible to strengthen this result and derive the convergence of the variance. Indeed, it is empirically observed that this quantity converges rapidly to a constant.

For a proper RL algorithm, it is clear that the standard deviation of $V_1^{\pi_k^i, M_k^i}(s_1)$ should shrink w.r.t. $k$. Otherwise, the algorithm fails to find out the optimal value and policy. Suppose that the standard deviation of $V_1^{\pi_k^i, M_k^i}$ is $\tilde{O}(k^{-1/\alpha})$, where $\tilde{O}$ is a variant of the big $O$ notation that ignores logarithmic factors. We need to set the scaling parameter $\lambda_k$ proportionally such that the variation of the log-posterior density term $\log \Pr(M_k^i|\mathcal{H}_k)$ matches that of the value term $V^{\pi_k^i, M_k^i}$. Given a specific model class and noise type, it is possible to derive a worst-case rate for the shrinkage of the value uncertainty and thus determine the proper value of $\lambda_k$. Nonetheless, the rate of information revealing could appear in an instance-dependent manner. That is to say, the optimal policy of some MDPs could be inherently easier to determine than the others. Even in a fixed MDP, the rate of uncertainty shrinkage could also change abruptly. For example, consider a ReLU bandit problem, where $B = \{a \in \mathbb{R}^d | \|a\|_2 \leq 1\}$, and the agent at each episode selects an action $a \in B$ and receives a Gaussian reward of mean $\max(\theta^\top a, 0)$. If it happens that the agent selects an action $a$ in the inactive region $I = \{a \in B | \theta^\top a \leq 0\}$, the resulting observation reveals little information about the optimal action. Suppose the parameter $\theta$ is selected such that the positive region $B \backslash I$ is of a maximum width $\epsilon > 0$. Then, in the worst case, the agent needs to explore $\Omega(1/\epsilon^{d-1})$ actions in order to find the positive region, but the uncertainty starts diminishing rapidly whenever the positive region is identified.

The view of matching the log probability variation with the value uncertainty provides rationales for the dynamic adjustment of $\lambda_k$. Meanwhile, it points out the potential limitation of decaying $\lambda_k$ with a fixed rate. In this paper, we focus on methods that scale $\lambda_k$ with a fixed rate, but as mentioned above, it would be fascinating to adjust it in an instance-dependent manner.

### 4.3   BOO Regret

In this section, we introduce the regret of BOO. The regret of BOO relies on both the decay rate of the scaling parameter and the complexity of model class. We assume that the model class resides in a $d$-dimensional RKHS, which roughly means any function in the model class can be represented as a linear function of a potentially unknown finite-dimensional feature map. This assumption is not very restrictive because it places no restriction on the choice of feature map except for the finitude of the dimension.

In Appendix B, we derive the asymptotic regret of the BOO algorithm in Theorem B.1. As suggested by Theorem B.1, the optimal asymptotic regret of BOO is $O\left(\sqrt{K} \log K\right)$, i.e., $\tilde{O}\left(\sqrt{K}\right)$, achieved by setting $\lambda_k^* = c/\sqrt{k}$, where $\lambda_k^*$ is the optimal scaling parameter and $c$ is a constant associated with the model class. More detailed derivations and conclusions can be found in Appendix B.

We can further interpret $\lambda_k^* = c/\sqrt{k}$ as $\lambda_k^* = \xi_k^V/\xi_k^M$, where $\xi_k^M$ stands for the variation of the log-posterior density for the model $M_k$, and $\xi_k^V$ represents the value uncertainty. In contrast to the discussion in Section 4, where we consider samples from the posterior distribution and the variation of log-posterior density remains constant, the width of the HPD region measured by the variation of log-posterior density is, in fact, growing at a rate depending on the log covering number. Accroding to Lemma D.7 in Appendix D.1, $\xi_k^M = O\left(\log k\right)$. Therefore, we have $\xi_k^V = \lambda_k^* \xi_k^M = O\left(\frac{\log k}{\sqrt{k}}\right)$. Section 5 will highlight the need of manipulating with $\xi_k^V$ and $\xi_k^M$ in optimization.

# 5 Optimization

In this section, we introduce optimization methods for BOO based on posterior sampling and gradient descent, respectively, and discuss in detail the problems of gradient-based optimization and our proposed solutions.

## 5.1 Optimization via Posterior Sampling

Algorithm 4 provides a method of optimizing the BOO objective via posterior sampling. As discussed previously, the optimal scaling parameter $\lambda_k$ for this algorithm is proportional to the value uncertainty, i.e., $\lambda_k = \xi_k^V = \lambda_k^* \xi_k^M$. The potential advantage of Algorithm 4 over PSRL is that, unlike PSRL, it does not require posterior samples to be independent. Even the requirement that samples are from the posterior can be relaxed. These features are essential either when the posterior distribution is approximated or when the posterior samples are produced by Markov chain Monte Carlo methods and are correlated.

## 5.2 Optimization via Gradient-Based Methods

We can also perform the optimization via gradient-based methods. The objective function of BOO consists of two parts, $V_1^{\pi, M_k^i}(s_1)$ and $\lambda_k(\log \Pr(\mathcal{H}_k | M_k^i) + \log \Pr(M_k^i))$. The gradients of the log-likelihood and the log-prior are easily obtained. For the value part, we provide a value model gradient that admits a similar form as the well-known policy gradient [36, 37] (proof in Appendix E.1). The value model gradient is amenable to Monte Carlo approximation and can be computed exactly for finite MDPs.

**Theorem 5.1** (Value model gradient). *Suppose that the transition function $P^{M_\theta}$ and reward function $R^{M_\theta}$ of model $M_\theta$, the gradient of the value $V_1^{\pi, M_\theta}(s_1)$ w.r.t. the model is*

$$\nabla_\theta V_1^{\pi, M_\theta}(s_1) = \mathbb{E}_{\tau \sim \pi, M_\theta} \left[ \sum_{h=1}^{H} \nabla_\theta \bar{R}^{M_\theta}(s_h, a_h) + \sum_{h=1}^{H-1} V_{h+1}^{\pi, M_\theta}(s_{h+1}) \nabla_\theta \log P^{M_\theta}(s_{h+1} | s_h, a_h) \right],$$
(4)

*where $\tau = (s_1, a_1, \ldots, s_H, a_H)$ is a trajectory, $\tau \sim \pi, M_\theta$ means that the trajectory is formed by the interaction of the policy $\pi$ and the model $M_\theta$, and $P^{M_\theta}(s_{h+1} | s_h, a_h)$ is the probability of $s_{h+1}$ under distribution $P^{M_\theta}(s_h, a_h)$.*

Nevertheless, the value model gradient suffers the same inefficiency just as the policy gradient [38] since the model gradient for a particular state-action pair is $0$ whenever it is not visited under the current model and policy.

We propose some techniques to improve the optimization efficiency of BOO, and conduct ablation experiments to verify the effectiveness of our proposed methods in Section G. Two of the most effective techniques are described below, and the rest of the techniques are detailed in Appendix F.

### 5.2.1 Mean Reward Bonus

In the gradient-based optimization, the model becomes optimistic on state-action pairs it visits frequently, which in turn makes these state-action pairs more appealing. This mutual strengthening phenomenon makes optimization easily stuck at local optima. One way to solve this problem is to increase the rewards of all state-action pairs, which raises the attractiveness of less visited state-action pairs. This can be achieved by adding a bonus term $\xi_k^V H \mathbb{E}_{(s,a) \sim U_{\mathcal{S} \times \mathcal{A}}}[R(s, a)]$ to the BOO objective. Here, $U_{\mathcal{S} \times \mathcal{A}}$ is the uniform distribution over the state-action space. The coefficient $\xi_k^V$ ensures that the bonus decays with the value uncertainty. The unvisited state-action pairs will eventually be visited by the policy because they have sufficiently high rewards. Our experiments will show that, in tabular setting, this method is very effective. However, a concern is that it might fail to scale to high dimensional state-action space.

### 5.2.2 Entropy Regularization

Another intricacy of the optimization is that the optimal solution of the BOO objective could change dramatically across the parameter space from episode to episode, which renders the optimization

extremely hard. A revealing fact is that a small change in the model could change the optimal policy dramatically, making the loss landscape unsmooth. Hence, we introduce an entropy-regularized optimization procedure, which starts with a high initial entropy regularization and gradually annealing. The entropy plays a role in smoothing the policy's loss landscape such that the optimal policy will not change drastically when the model changes. The smoothing effect of entropy regularization is also discussed previously in [39].

The entropy-regularized learning objective mimics the maximum entropy RL [40]:

$$\tilde{J}_k = V_1^{\pi,M}(s_1) + \lambda_k \log \Pr(M|\mathcal{H}_k) - \xi_k^V \zeta \, \mathbb{E}_{\tau \sim \pi, M} \left[ \sum_{h=1}^{H} \mathrm{KL}(\pi(s_h, h) \| \hat{\pi}(s_h, h)) \right]$$

$$= \tilde{V}_1^{\pi,M}(s_1) + \lambda_k \log \Pr(M|\mathcal{H}_k),$$

(5)

where $\mathrm{KL}(p\|q)$ stands for the relative entropy, $\hat{\pi}$ is a prior policy ensuring $\pi$ is absolutely continuous w.r.t. $\hat{\pi}$, the hyperparameter $\zeta$ controls the amount of entropy, and $\tilde{V}$ is the entropy-regularized value.

The entropy term is downscaled in proportion to the value uncertainty such that the influence of regularization diminishes with time. We denote by $\mathcal{M}_\theta$ the set of models parameterized by $\theta$. Let $b_\epsilon$ be the smallest number ensuring that, for any $M_\theta \in \mathcal{M}_\theta$, there exists $\pi_\epsilon \in \Pi_\epsilon = \{\pi \mid \mathrm{KL}(\pi(s,h)\|\hat{\pi}(s,h)) \leq b_\epsilon, \forall s \in \mathcal{S}, h \in [H]\}$ such that $\sup_{\pi^*} V_1^{\pi^*,M}(s_1) - V_1^{\pi_\epsilon,M}(s_1) \leq \epsilon$. Then, the value of the maximum entropy policy $\tilde{\pi}^* = \arg\max_{\tilde{\pi}} V_1^{\tilde{\pi},M}(s_1)$ satisfies that

$$V_1^{\tilde{\pi}^*,M}(s_1) \geq V_1^{\tilde{\pi}^*,M}(s_1) - \xi_k^V \zeta \, \mathbb{E}_{\tau \sim \tilde{\pi}^*, M} \left[ \sum_{h=1}^{H} \mathrm{KL}(\tilde{\pi}^*(s_h, h) \| \hat{\pi}(s_h, h)) \right]$$

$$\geq V_1^{\pi_\epsilon,M}(s_1) - \xi_k^V \zeta \, \mathbb{E}_{\tau \sim \pi_\epsilon, M} \left[ \sum_{h=1}^{H} \mathrm{KL}(\pi^\epsilon(s_h, h) \| \hat{\pi}(s_h, h)) \right]$$

$$\geq \sup_{\pi^*} V_1^{\pi^*,M}(s_1) - \epsilon - \xi_k^V \zeta b_\epsilon.$$

(6)

If $\epsilon$ is sufficiently small, then the sub-optimality gap caused by entropy regularization will decrease at the same rate as the decay of value uncertainty, which ensures that the resulting policy covers multiple uncertain actions without detriment to the performance.

The entropy-regularized value is equivalently defined by the following Bellman backup,

$$\tilde{V}_h^{\pi,M}(s_h) = \mathbb{E}_{a_h \sim \pi(s_h, h)}[\tilde{Q}_h^{\pi,M}(s_h, a_h)] - \xi_k^V \zeta \mathrm{KL}(\pi(s_h, h) \| \hat{\pi}(s_h, h)), \, \forall h \in [H],$$

$$\tilde{Q}_h^{\pi,M}(s_h, a_h) = \bar{R}^M(s_h, a_h) + \mathbb{E}_{s_{h+1} \sim P^M(s_h, a_h)} \left[ \tilde{V}_{h+1}^{\pi,M}(s_{h+1}) \right], \, \forall h \in [H-1],$$

(7)

where $\tilde{Q}_H^{\pi,M}(s_H, a_H) = \bar{R}^M(s_H, a_H)$. The optimization of the policy can be carried out by, for example, the maximum entropy actor-critic algorithm [40] or the soft actor-critic algorithm [41]. In finite MDPs, the optimal value is given by soft value iteration,

$$\tilde{V}_h^{\tilde{\pi}^*,M}(s_h) = \xi_k^V \zeta \log \sum_{a_h \in \mathcal{A}} \hat{\pi}(a_h|s_h, h) \exp \left( \frac{\tilde{Q}_h^{\tilde{\pi}^*,M}(s_h, a_h)}{\xi_k^V \zeta} \right),$$

(8)

where $\pi(a_h|s_h, h)$ is the probability of $a_h$ under the distribution $\pi(s_h, h)$. The maximum entropy policy $\tilde{\pi}^*$ is then derived from the optimal value as

$$\tilde{\pi}^*(a_h|s_h, h) = \hat{\pi}(a_h|s_h, h) \exp \left( \frac{\tilde{Q}_h^{\tilde{\pi}^*,M}(s_h, a_h) - \tilde{V}_h^{\tilde{\pi}^*,M}(s_h)}{\xi_k^V \zeta} \right).$$

(9)

The gradient of the entropy-regularized value w.r.t. the model is similar to that specified in Theorem 5.1, with the value replaced by the entropy-regularized value.

## 6 Experiments

This section compares the performance of BOO with PSRL and UCRL2 in RiverSwim, Chain, and Random MDPs. RiverSwim and Chain are hard-exploration MDPs requiring the agent to explore

efficiently, while Random MDPs test the average performance. Two implementations of BOO are presented, namely, FiniteBOO and BPS. FiniteBOO is the BOO with entropy regularization and mean reward bonus mentioned in Section 5.2, which is empirically found to be the best variant in tabular setting according to the ablation study. BPS is an implementation of Algorithm 4 (BOO via posterior sampling).

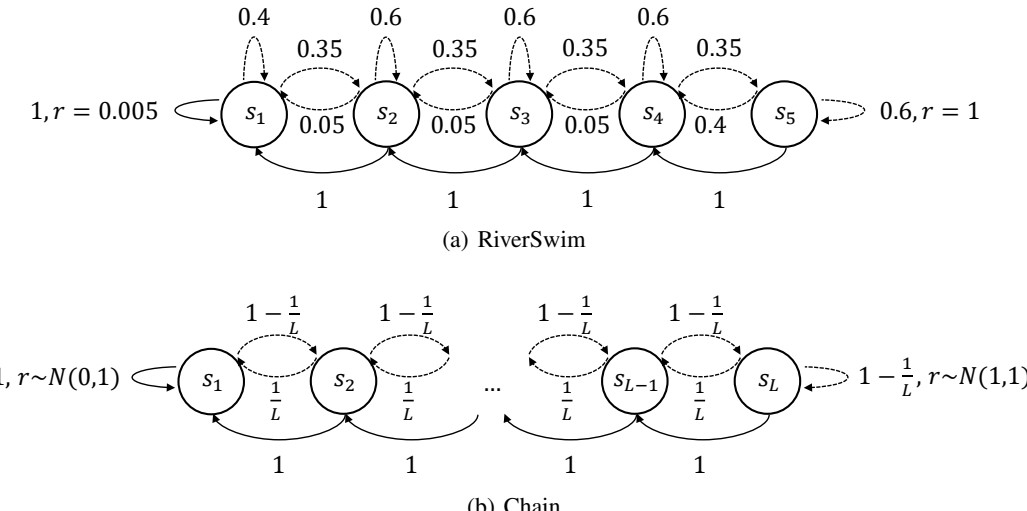

(a) RiverSwim

(b) Chain

Figure 1: Illustrative diagrams for RiverSwim and Chain. Solid and dotted arrows represent actions, "left" and "right", respectively, with the transition probability tagged. The action left never fails, but the action right often fails. Except for the rewards shown in the figure, the rewards of other state-action pairs are all zero.

## 6.1 RiverSwim

The RiverSwim is an MDP where states are organized in chains, and the agent can move left or right, as shown in Figure 1(a). Although the rightmost state has a huge reward, the action of moving right fails with a high probability. Only a policy moves right at each time period has a small chance of success.

We start with the experiment on the RiverSwim to demonstrate the performance of our algorithm in the face of high transition uncertainty. We perform experiments for ten seeds on RiverSwim with $|\mathcal{S}| = H = 5, |\mathcal{A}| = 2$ and record the cumulative regret over 100,000 time steps. As shown in Figure 2(a), our algorithm is competitive to PSRL and outperforms UCRL2 significantly.

## 6.2 Chain

The chain MDP is a variant of the RiverSwim, which has Gaussian rewards and relatively deterministic transitions, as shown in Figure 1(b). Although transitions are relatively certain, the stochastic rewards make the problem difficult to explore. The horizon $H$ and the number of states $|\mathcal{S}|$ are equal to the length of the chain.

We evaluate our algorithms in Chain of a length $L \in \{10, 20, 40\}$ for $100,000$ episodes and ten random trials. Figure 2(d~f) illustrates that our algorithm compares favorably with PSRL and UCRL2, which certifies the effectiveness of BOO in problems requiring long-term planning.

## 6.3 Random MDPs

Random MDPs are tabular MDP models randomly generated from a prior distribution and used to test the general performance of the algorithm.

We randomly generate 100 stochastic MDPs with $|\mathcal{S}| = |\mathcal{A}| = H = 5$ and $|\mathcal{S}| = H = 20, |\mathcal{A}| = 5$ from the prior and measure the performance of algorithms over $10,000$ timesteps. As shown in

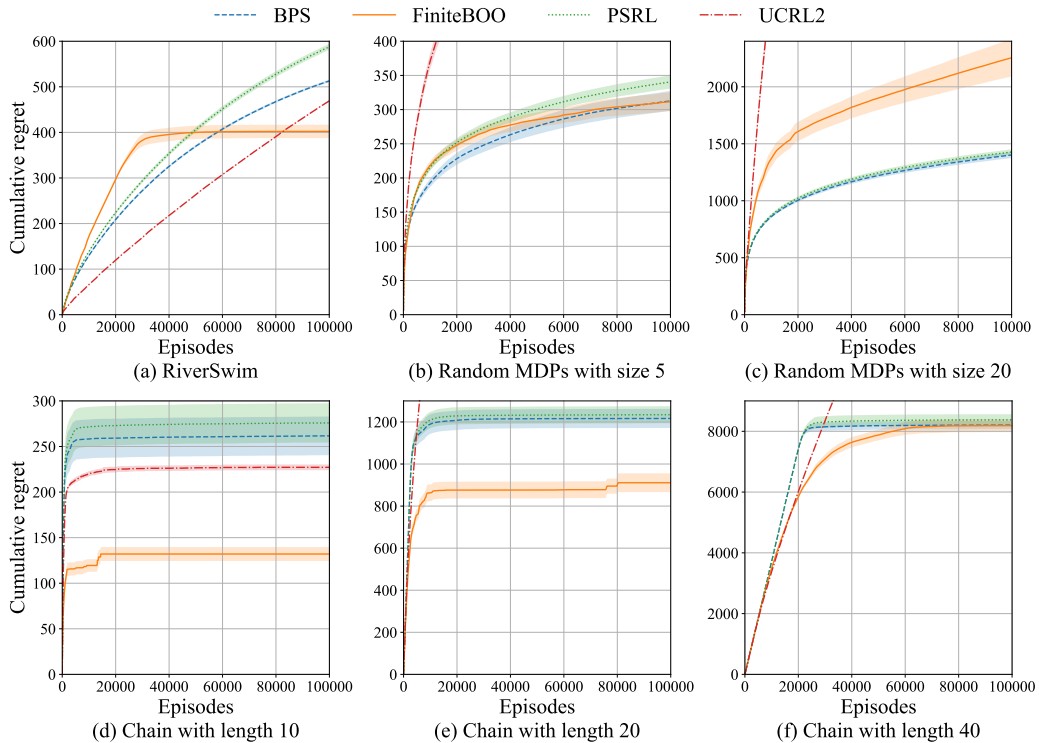

Figure 2: (a∼f) show the cumulative regrets of different algorithms on RiverSwim ($|\mathcal{S}| = H = 5, |\mathcal{A}| = 2$), Random MDPs ($|\mathcal{S}| = H = 5, |\mathcal{A}| = 5$), Random MDPs ($|\mathcal{S}| = H = 20, |\mathcal{A}| = 5$), and the chains of different lengths, respectively.

Figure 2(b), the performance of BOO and BPS is close to PSRL, and both algorithms outperform UCRL2 significantly. However, the performance gap between FiniteBOO and PSRL in Figure 2(c) indicates that there remains a challenge in the optimization of large-scale MDPs.

## 7   Conclusion

This paper proposes BOO as a generic model-based RL algorithm. It is provably sample-efficient and enjoys $\tilde{O}(\sqrt{K})$ regret for models in a finite-dimensional RKHS, where $K$ is the number of episodes. To optimize the BOO objective, we propose the value model gradient and optimization techniques, such as entropy regularization, to improve its efficiency. Through our experiments, we have shown that BOO is competitive with PSRL and outperforms UCRL2 greatly. However, to apply BOO in real-world RL problems, there remains lots of work to be done. Importantly, we need to develop methods that further facilitate the gradient-based optimization of BOO in large-scale problems. It is also an appealing direction for future work to adapt the scaling parameter of BOO on an instance-dependent basis.

## Acknowledgements

This work is supported by National Key Research and Development Program of China (2020AAA 0107200), the National Science Foundation of China (61921006, 61876119, 62276126), the Natural Science Foundation of Jiangsu (BK20221442), and the Fundamental Research Funds for the Central Universities (022114380010).

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
