# OpenReview forum: "Bayesian Optimistic Optimization: Optimistic Exploration for Model-based Reinforcement Learning"
_NeurIPS.cc/2022/Conference — NeurIPS 2022 Accept_

### Official Review · Reviewer_pz9j · 2022-07-08

**Rating:** 6
**Confidence:** 3
**Soundness:** 3 good
**Presentation:** 2 fair
**Contribution:** 3 good

**Summary:**

The paper considers the exploration-exploitation tradeoff in reinforcement learning and develops a new algorithm rooted in the "optimisim in the face of uncertainty" principle. The paper starts from a Bayesian perspective where the policy is selected optimistically w.r.t. a credible, high-probability region of the model posterior. It then develops a lagrangian relaxation (BOO) and provides a regret bound that is sublinear depending on a schedule for the Lagrange multiplier and novel upper-bounds for the distributional eluder dimension. The paper also shows that BOO can be approximated by optimizing over samples from the posterior and, based on that, developes a practical algorithm for parametric models based on the reinforce trick applied to the model's stochasticity. It also extends this methods with the natural policy gradient. The main paper does not provide any experimental evaluation, though a small-scale study is included in the appendix.

**Questions:**

Could you summarize what you view as the main technical novelties in the derivations of the regret bound?

It seems to me that the policy gradient in (8) would, like standard reinforce, yield high-variance results for close-to-detministic models as the gradient of the log-probability becomes very steep. Usually that is "resolved" by adding an entropy bonus for the policy, but this seems unreasonable for the model. How do you deal with this in practice?

The Related work section is currently a bit sparse and should be expanded. Below are some recent papers on OFU that come to mind, it would probably be good to discuss them in relation to the current paper.
- Chowdhury et. al, "Online Learning in Kernelized Markov Decision Processes", AISTATS, 2019
- Kakade et. al, "Information theoretic regret bounds for online nonlinear control", NeurIPS 2020
- Curi et. al., Efficient Model-Based Reinforcement Learning through Optimistic Policy Search and Planning, NeurIPS 2020

**Limitations:**

For the theoretical part are discussed throughout the paper, but not for Sec. 6

**Strengths And Weaknesses:**

The paper investigates the relevant topic of efficient exploration in reinforcement learning. It provides novel regret bounds. As far as I can tell the theoretical analysis is rigorous and presents an interesting perspective on how to side-step defining explicit confidence regions for the model. The main limitation, which is also discussed in the paper, is that this comes at the cost of having to manually specify a Lagrange multiplier to trade-off model-likelihood v.s. gained value. While the paper provides regret bounds for fixed schedules that depend on the model-class, without an adaptive selection this severely limits the usefullness of the approach.

The paper tries to cover both the theoretical contribution and the practical question of how to optimize it. While this is commendable in general, in this case experiments are banished to the appendix. In it's current form, imho anything beyond Theorem 6.1 does not add significant value, since empirical concerns like entropy-regularization would also require empirical evaluation. The current experiments in the appendix lack significant details, but are limited to toy systems and do not seem to cover any results from Sec. 6.

---

> ### Author Response · Authors · 2022-08-02
> **Response to Reviewer pz9j**
>
> We are grateful for your careful reading and prized comments. We answer your concerns about the related work and empirical evaluation in the general response. The responses to other questions are listed below.
>
> **Q1**: Technical novelties in the derivation of the regret bound.
>
> **A1**: The main technical novelty is that instead of bounding the concentration regret via the width of the confidence set as in previous work, we show that the number of large per-episode concentration regret is small and express the concentration regret as a novel Riemann-Stieltjes integral.
>
> **Q2**: How to reduce the variance of the model gradient?
>
> **A2**: Similar to the classic policy gradient, we can subtract a baseline to reduce the variance of the model gradient. With regard to the gradient variance of the close-to-deterministic models, we believe this variance is inherent to the statistical model and is characterized by the Fisher information matrix at the real model parameter. Therefore, we need to properly parameterize the model so as to avoid a large variance.

---

### Official Review · Reviewer_hRsL · 2022-07-12

**Rating:** 6
**Confidence:** 2
**Soundness:** 3 good
**Presentation:** 3 good
**Contribution:** 3 good

**Summary:**

This paper proposes a new model-based RL that exploits the principle of optimism in the face of uncertainty (OFU). The proposed algorithm, Bayesian optimistic optimization (BOO) is shown to be sample-efficient and less computationally expensive than previous Bayes-adaptive MDP methods. The paper also provide theoretical analysis to show that the algorithm has a sublinear regret for many model classes.

**Questions:**

See below

**Limitations:**


The paper proposes an interesting idea to combine OFU into MBRL. Constructing an uncertainty constraint is not new in MBRL. The step-by-step derivation from start to the final algorithm CBOO is very clear and easy to follow. The use of Lagrangian Relaxation of CBOO is sensible to exploit uncertainty in making decisions. I was wondering how this is different from adding UCB, ie. probably the closest work is [1]. Does it lead to a totally different behavior for optimization/online planning? Can the authors also comment or discuss about your approach in comparisons to previous work that already introduced uncertainty bonus to Bayes-Adaptive MDP?

The theoretical result reported in Theorem 4.1 is interesting. Can the authors provide a bit of insights about it, i.e. on why the variance is at least m/2 is good thing for RL?

It is also very interesting to see the authors provided practical optimization aspects for the proposed algorithm. It would be very interesting if this can be realized with empirical results.

[1] Arthur Guez, Nicolas Heess, David Silver, Peter Dayan:
Bayes-Adaptive Simulation-based Search with Value Function Approximation. NIPS 2014: 451-459



**Strengths And Weaknesses:**

Strength:
- Principled method to integrate OFU into MBRL.
- Theoretical analysis showing a sublinear regret
- An optimization based on the natural gradient and entropy
regularization

Weakness:
- No empirical results
- Lacking discussions on related work of OFU on Batesian BMRL.

---

> ### Author Response · Authors · 2022-08-02
> **Response to Reviewer hRsL**
>
> Thank you for the careful evaluation. We address your concerns about the empirical results and the related work in the general response to all reviewers and answer your questions in the following part.
>
> **Q1**: What is the connection and difference to previous work based on Bayes-Adaptive MDPs (BAMDPs)?
>
> **A1**: As mentioned in the general response, we believe methods based on BAMDPs are entirely different from OFU algorithms since they try to learn a policy (or a value function) that reacts to belief distributions of MDPs or histories of past actions and observations. In terms of optimization/online planning, methods based on BAMDPs generally rely on online planning since dynamic programming or approximate dynamic programming is rarely possible for these methods, and OFU algorithms often have broader choices. We are not sure about what ‘uncertainty bonus’ means since we failed to find any in the referred paper. In the area of Bayesian reinforcement learning, there is indeed some work [1,2] based on the exploration bonus, but, to the best of our knowledge, these methods require access to the model posterior distribution and are restricted to the tabular or finite MDPs.
>
> **Q2**: Can the authors provide a bit of insights about the Theorem 4.1, i.e., on why the variance is at least m/2 is good thing for RL?
>
> **A2**: This result is a direct consequence of the asymptotic normality of the posterior distribution because an $m$-dimensional normal distribution has m/2 variance. It is not involved in the regret analysis and is only used to derive the scaling parameter for Algorithm 4 (BOO via posterior sampling). Hence, the m/2 variance is neither a good thing nor a bad thing for RL.
>
> **References:**
> [1] J. Z. Kolter and A. Y. Ng, “Near-Bayesian exploration in polynomial time,” in Proceedings of the 26th Annual International Conference on Machine Learning, New York, NY, USA, Jun. 2009, pp. 513–520. doi: 10.1145/1553374.1553441.
> [2] J. Sorg, S. Singh, and R. L. Lewis, “Variance-based rewards for approximate Bayesian reinforcement learning,” in Proceedings of the Twenty-Sixth Conference on Uncertainty in Artificial Intelligence, Arlington, Virginia, USA, Jul. 2010, pp. 564–571.

---

### Official Review · Reviewer_EVFD · 2022-07-17

**Rating:** 5
**Confidence:** 2
**Soundness:** 3 good
**Presentation:** 2 fair
**Contribution:** 2 fair

**Summary:**

This paper presents a model-based reinforcement learning algorithms based on the optimism in the face of uncertainty. First of all, the algorithm CBOO is constructed based on Bayesian credible region. However, CBOO is difficult to use in practice and therefore, the authors apply Lagrangian relaxation and posterior sampling to construct practical algorithm. Then, the authors provide regret analysis and conditions for various model setting. Finally, the authors discuss the optimization of the BOO.

**Questions:**

1. This paper seems to deal with frequentist RL problem. But the authors introduce the definition of Bayesian regret in preliminaries. This confuses me if I'm understanding correctly. Would you clarify this part?

2. Many previous frequentist model-based RL algorithms also uses similar concept with Bayesian credible region. For example, in [1], UCRL2 also introduces ‘plausible MDPs’ $\mathcal{M}_k$, where true MDP $M$ is in $\mathcal{M}_k$ with high probability $1-\delta$. Then, UCRL2 choose optimistic MDP $\tilde{M}_k$ among these plausible MDPs. I would like to know the main difference between BOO and previous model-based RL algorithms in finite MDP setting. It will be very helpful for understanding this paper.

3. How can we construct Algorithm 4 in practice? In my opinion, Algorithm 2 is nothing new and already well-known part. However, as the authors mentioned in Section 4.1, obtaining optimistic MDP (from credible region), is not trivial. Previous model-based RL algorithms have focused on how to make it possible in practice. Then, how can we construct a practical OFU algorithm based on Algorithm 4?

4. In conclusion section, the authors claimed BOO enjoys sublinear regret. Where can we know that the regret is sublinear? In Theorem 5.1, (7) is the regret bound for the best $v$. But, the order of $K$ is $1+\text{(some positive)}\ge 1$: as the number of episode increases, the regret bounds exponentially increases. It looks too weird to me. Would you please correct my misunderstanding?

[1] Auer, Peter, Thomas Jaksch, and Ronald Ortner. "Near-optimal regret bounds for reinforcement learning." NIPS 2008.


**Ethics Review Area:**

["I don’t know"]

**Limitations:**

Yes

**Strengths And Weaknesses:**

**Strengths**:

The proposed algorithm is a general algorithm for various model classes. In addition, the authors provide regret analysis to support the proposed algorithm.

**Weakness**:

1. Related work.

There are many model-based RL algorithms in finite MDPs. However, this paper presents only a very brief introduction to the related work. Because BOO can be reduced to finite MDPs, comparisons with previous works would be very helpful for understanding (e.g. connections between Theorem 5.1 in this paper and regret analysis in previous work)

2. Clarity

In this paper, notations are very confused. For example, $M$ is used to express MDP and to define covering number. Also, there is no definition of $\mathcal{X}$ in the paper. Finally, what is the definition of model – only transition, or only policy, or both? It is very confusing because there are no connection between $\mathcal{X}$, $\mathcal{F}$ and notations in MDP.

3. Implementation

It is not clear how to implement practical algorithm. It would be good if the authors could suggest at least the algorithms for models in Table 1. (how to compute & update posterior, how to sample MDPs from posterior, how to compute $\arg\max_\pi$, etc.)

---

> ### Author Response · Authors · 2022-08-02
> **Response to Reviewer EVFD**
>
> Thanks for your invaluable comments. The related work and the implementation issue are addressed in the general response. In terms of clarity, the definition of $\mathcal{X}$ and  $\mathcal{F}$ and their connection to notations of MDPs are specified in Assumption 1. Throughout the paper, we use the terms model and MDP interchangeably. We will try to optimize our presentation and resolve these unclarities in the revised version. In the following, we will answer your questions one by one.
>
> **Q1**: Why do you introduce the definition of Bayesian regret?
>
> **A1**: The Bayesian regret is introduced mainly for motivating the design of the algorithm. Concretely, it is used in Section 4.1 for measuring the performance of the CBOO algorithm and motivates the use of the HPD region.
>
> **Q2**: What is the difference between BOO and previous model-based RL algorithms in finite MDP settings?
>
> **A2**: We take UCRL2 as an example. UCRL2 utilizes the structure of the tabular MDP to construct the confidence set. Specifically, it builds factorized confidence sets for different state-action pairs separately. This allows efficient computation at the cost of generality and some statistical efficiency. On the contrary, CBOO and BOO are based on a credible region for the entire MDP, which have higher computational overhead when specialized to tabular MDPs but achieve better performance and are more general.
>
> **Q3**: How can we construct Algorithm 4 in practice?
>
> **A3**: Algorithm 4 (BOO via posterior sampling), by itself, is an implementation of Algorithm 3 (BOO). It is easily implementable when the transition and reward distribution admits a conjugate prior, in which case the posterior distribution has a closed form, and posterior samples are readily available. For example, in tabular MDPs, the posterior distribution of the transition function is a Dirichlet distribution when we have a Dirichlet prior over the transition parameters. Another example is that the posterior distribution of the transition function is a Gaussian process when the transition has a Gaussian noise and we have a Gaussian process prior over it. For models without a conjugate prior, we can approximate the model posterior via any Bayesian inference method or sample from the posterior via the MCMC techniques.
>
> **Q4**: Where can we know that the regret is sublinear?
>
> **A4**: We apologize for the unclarity and will modify the theorem to make it more accessible. At the optimal choice of the scaling parameter, it can be shown that the regret is at an order of $\tilde O\left(K^{1-1/((c_1+1)(e_1+2))}\right)$, where $c_1,e_1\geq0$. When specialized to the finite-spectrum RKHS, it recovers the best possible $\tilde O(\sqrt{K})$ regret.

---

> > ### Comment · Reviewer_EVFD · 2022-08-09
> > **Response to the authors**
> >
> > Thanks to the authors for their responses.
> >
> > I checked the revision, especially Section 5 (including Theorem 5.1).
> >
> > The major concern (Theorem 5.1) has been addressed.
> >
> > I change the score from 4 to 5.

---

> ### Author Response · Authors · 2022-08-09
> **Sincerely looking for further feedback**
>
> Dear Reviewer EVFD,
>
> Thanks again for your comments. We hope our response clarifies your concerns. Since the author-reviewer discussion will soon end, please let us know if you have any other questions. If there are no more questions, we would appreciate it if you could kindly raise the score.
>
> Sincerely yours,
> Authors of Paper10202

---

### Author Response · Authors · 2022-08-02
**General Response**

We thank the careful and valuable comments from all reviewers. Some common issues will be addressed here, with others replied respectively.

### 1. Related Work

We agree that our discussion on the related work is insufficient and will add a thorough discussion in the revised version. Here, we provide a discussion on papers mentioned by reviewers.

One line of research resolves the exploration and exploitation dilemma by formalizing the reinforcement learning problem as a planning problem in Bayes-Adaptive MDPs (BAMDPs), which treats the unknown MDP parameter as an additional hidden variable and maintains a belief distribution of the parameter. [1] proposes an efficient planning algorithm in BAMDPs, which only requires sampling from the model posterior distribution. However, this line of work shares the scalability problem, which is caused by the exponential increase of possible histories w.r.t. the planning horizon. That is, the planning in BAMDPs is PSPACE-complete [2], meaning that it demands exponential time and space to solve. Intuitively, this is because methods based on BAMDPs deal with the belief distributions of MDPs (or histories) rather than a single MDP.

Methods following the OFU principle construct an optimistic estimate of the unknown MDP and execute its optimal policy. Since these methods plan on a single MDP estimate, they are computationally preferable compared to methods based on BAMDPs. The UCRL2 is one such method. However, unlike tabular and linear MDPs for which constructing an optimistic estimate is analytically tractable, constructing an optimistic estimate for general MDPs involves a constrained joint optimization of model and policy and is computationally prohibitive. Hence, previous model-based OFU algorithms [4,5] for general model classes cannot be implemented and rely on posterior sampling for exploration.

The posterior sampling for reinforcement learning (PSRL) is argued [6] to be better than optimism since it is more computationally tractable. Nevertheless, as opposed to OFU algorithms, incremental implementation of PSRL is challenging, preventing it from scaling up. [7] was devoted to resolving the intractability of optimism for general models. It proposes to convert the joint optimization of model and policy into a hallucinated control problem. This approach ignores the correlation between state-action pairs and treats them separately, causing inefficiency as reflected by the extra dependency on the cumulative posterior variance in their regret bound. Our methods avoid this since maximizing the log-posterior density naturally enforces the correlation between state-action pairs.

Previous regret analysis [4,5] for general model classes relies on the fact that the constructed model is the most optimistic in the confidence set. However, our algorithm, BOO, does not satisfy this requirement. In fact, the model constructed by BOO may not even belong to a confidence set or a credible region. This difference entails a distinct analysis, where we show that neither the large deviation from the real model nor the possibly pessimistic estimation of the model causes a large regret.

### 2. Empirical Evaluation

In the Appendix, we provided a brief empirical evaluation. Now, we upload a new appendix, which incorporates an ablation study of techniques in Section 6 and some new results, along with all experimental details. The ablation study shows clearly that the proposed optimization techniques are indeed effective. In light of the performance, BOO could outperform PSRL when the optimization is done properly and performs reasonably well in most experiments. However, the gradient-based optimization of BOO in large-scale MDPs remains challenging even with the proposed methods and awaits further investigation.

 **References**:

[1] A. Guez, N. Heess, D. Silver, and P. Dayan, “Bayes-Adaptive Simulation-based Search with Value Function Approximation,” NIPS 2014

[2] C. H. Papadimitriou and J. N. Tsitsiklis, “The Complexity of Markov Decision Processes,” Mathematics of OR, vol. 12, no. 3, pp. 441–450, Aug. 1987, doi: 10.1287/moor.12.3.441.

[3] T. Jaksch, R. Ortner, and P. Auer, “Near-optimal Regret Bounds for Reinforcement Learning,” J. Mach. Learn. Res., vol. 11, pp. 1563–1600, Aug. 2010.

[4] S. R. Chowdhury and A. Gopalan, “Online Learning in Kernelized Markov Decision Processes,” AISTATS 2019

[5] S. Kakade, A. Krishnamurthy, K. Lowrey, M. Ohnishi, and W. Sun, “Information Theoretic Regret Bounds for Online Nonlinear Control,” NeurIPS 2020

[6] I. Osband and B. V. Roy, “Why is Posterior Sampling Better than Optimism for Reinforcement Learning?,” ICML 2017

[7] S. Curi, F. Berkenkamp, and A. Krause, “Efficient Model-Based Reinforcement Learning through Optimistic Policy Search and Planning,” in NeurIPS 2020

---

### Meta-Review · Area_Chair_JAtb · 2022-08-26

**Recommendation:** Accept
**Confidence:** Less certain

**Metareview:**

The paper presents a new model-based approach based on optimism in the face of uncertainty.
After reading each other's reviews and the authors' feedback, most of the reviewers' concerns were solved and the reviewers agree that this paper deserves publication.
However, while preparing the final version of their paper, the authors have to consider the reviewers' suggestions and in particular, they are expected to extend the related work discussion and move some of the experimental results from the appendix to the main paper.

**Award:**

No

---

### Decision · Program_Chairs · 2022-09-14

Accept